# Assigning the Origin of Microbial Natural Products by Chemical Space Map and Machine Learning

**DOI:** 10.3390/biom10101385

**Published:** 2020-09-28

**Authors:** Alice Capecchi, Jean-Louis Reymond

**Affiliations:** Department of Chemistry and Biochemistry, University of Bern, Freiestrasse 3, 3012 Bern, Switzerland; alice.capecchi@dcb.unibe.ch

**Keywords:** natural products, databases, cheminformatics, chemical space, visualization, molecular fingerprints, machine learning, support vector machine, origin classification

## Abstract

Microbial natural products (NPs) are an important source of drugs, however, their structural diversity remains poorly understood. Here we used our recently reported MinHashed Atom Pair fingerprint with diameter of four bonds (MAP4), a fingerprint suitable for molecules across very different sizes, to analyze the Natural Products Atlas (NPAtlas), a database of 25,523 NPs of bacterial or fungal origin. To visualize NPAtlas by MAP4 similarity, we used the dimensionality reduction method tree map (TMAP). The resulting interactive map organizes molecules by physico-chemical properties and compound families such as peptides and glycosides. Remarkably, the map separates bacterial and fungal NPs from one another, revealing that these two compound families are intrinsically different despite their related biosynthetic pathways. We used these differences to train a machine learning model capable of distinguishing between NPs of bacterial or fungal origin.

## 1. Introduction

Natural products (NPs) of microbial origin are an important source of drugs. Numerous examples of antibiotic, antifungal, immunosuppressive, anti-inflammatory, and anti-cancer agents on the market originate from fungi or bacteria [1]. A notable effort has been made to explore the known and virtual chemical space of microbial NPs and NPs in general [2,3,4,5]. Furthermore, machine learning (ML) has been extensively applied to natural product structures, for example, to classify limonoids and protolimonoids [6], to establish the structural class of a natural product with its NMR data [7], to learn estimates of natural product conformational energies [8], to generate derivates of NPs or compounds with natural product characteristics [9,10,11], to predict meridian in Chinese traditional medicine [12], and to elucidate the biological effects of natural products [13]. The recently published Natural Products Atlas (NPAtlas) is a collection of 25,523 NPs of fungal and bacterial origin [14]. Among other tools, the NPAtlas website (https://www.npatlas.org/joomla/) provides a global view of the database in a spherical representation. To generate this view, the NPAtlas entries are clustered by Dice similarity [15] using the substructure fingerprint ECFP4 (an extended connectivity fingerprint with a diameter of four bonds) [16]. The resulting clusters are grouped in nodes, which are arranged in a spherical plot where the position of each node is determined by molecular formulas. While this representation provides interesting insights into the composition of the NPAtlas, individual compounds cannot be visualized in the global overview but only within clusters, therefore, comparing compounds across two different clusters is not possible.

A defining feature of NPAtlas is that NPs featured in this database span across a broad range of sizes, with the largest NPs reaching up to almost 3 kDa (Appendix A). We showed recently that the ECFP4 fingerprint, although well suited for small molecule drugs, performed poorly with larger molecules typically found in NP collections such as lipids, oligosaccharides, and peptides [17]. To address this limitation, we recently investigated molecular fingerprints combining the concept of atom pairs [18], which is well suited to analyze large molecules such as proteins and peptides [19,20,21,22], with extended connectivity substructures and bit compression using MinHash as used in the substructure fingerprint MHFP6 [23], and proposed the MinHashed atom pair fingerprint with a diameter of four bonds (MAP4) as an optimal molecular fingerprint to analyze molecules of very different sizes [17].

Here we asked the question of whether analyzing NPAtlas using MAP4 might provide new insights into the composition of this collection. To organize molecules according to their MAP4 similarity, we used TMAP, a recently reported dimensionality reduction method suitable to analyze very large high-dimensional datasets [24]. TMAP performs better for the visualization of large high-dimensional data sets than other dimensionality reduction methods such as t-SNE [25] or UMAP [26]. Furthermore, TMAP is particularly well suited to analyze databases of molecules associated with MinHashed fingerprints. 

## 2. Materials and Methods

### 2.1. NPAtlas Dataset

The December 2019 version of the NPAtlas was used. This version of the database contains 25,523 entries, 15,759 of fungal origin, and 9764 entries of bacterial origin, with no entry sharing bacterial and fungal origin. For each compound, a simplified molecular-input line-entry system (SMILES), molecular weight (MW), origin (fungal or bacterial), and the DOI of the associated publication were downloaded. For the MAP4 fingerprint calculation, the SMILES were canonicalized [27] and the stereochemistry was removed using the RDKit toolkit [28]. After removing stereochemistry, the NPAtlas counts 23,928 unique SMILES and 76 entries common among both origins.

### 2.2. MAP4 Fingerprint

The MAP4 fingerprint combines the circular substructure and atom pair fingerprints concepts. MAP4 encodes each atom pair in a molecule as the SMILES of the circular substructure of radii 1 and 2 around both atoms and the distance in bonds that separates them. The resulting set of strings is hashed to integers using the SHA-1 algorithm [29] and MinHash scheme [30]. The obtained MAP4 fingerprint is an array of unsorted numbers, where each feature is characterized by its value and its position in the array (index). MAP4 perceives substructure details while maintaining a global overview; therefore, it is suitable to describe molecular structures across different sizes. The similarity between two MAP4 fingerprints a and b was calculated: (1) counting of elements with the same value and the same index across a and b, and (2) dividing the obtained value by the number of elements of fingerprint a. The similarity between two MinHashed MAP4 fingerprints calculated as described above is an estimation of the Jaccard Similarity between the two non-MinHashed objects [30]. For a detailed explanation if the MAP4 implementation and benchmark, please refer to our recent publication [17]. The 1024-dimensions MAP4 fingerprint of all NPAtlas entries was calculated using canonical SMILES without stereochemistry information.

### 2.3. TMAP Layout

The TMAP layout was calculated from the MAP4 fingerprint dataset using the open-source implementation of TMAP [24]. In short, the indices generated by the MinHash procedure of the MAP4 calculation were used to create a locality-sensitive hashing (LSH) forest [31] of n trees. For each NPAtlas entry, the *k* approximate nearest neighbors (NNs) in the MAP4 feature space are then extracted from the LSH forest to form a graph in which nodes are the structures and edges are the NN relationships weighted by the fingerprint distance. The Kruskal’s algorithm was then applied to remove cycles and to find the path with the lowest total distance between all molecules in the graph [32]. Finally, Fearun [33] was used to interactively display the obtained minimum spanning tree. In this study, we set n = 32 and *k* = 20.

### 2.4. Properties Calculation

For all NPAtlas entries, the number of hydrogen bond acceptors (HBA) and hydrogen bond donors (HBD), logP following Crippens approach (AlogP) [34], topological polar surface area (TPSA), and fraction of sp3 carbon (fsp3C) were calculated with RDKit. The boiling point was calculated using the open-source code of the JRgui [35] as the Joback boiling temperature [T_Job_, Equation (1)] [36],
(1)$$T_{Job}=198.2+\underset{i}{\sum }N_{i}t_{bi}$$
where *N*_*i*_ is the occurrence of a functional group in the molecule, and *t*_*bi*_ is its empirically obtained contribution value. Molecules that violated more than one Lipinski rules [37] were labeled as non-Lipinski. To identify glycosylated and/or peptidic structures, Daylight [38] SMARTS language was used. SMILES arbitrary target specification (SMARTS) were used with RDKit to identify NPAtlas entries containing a dipeptide substructure, defined as “[NX3,NX4+][CH1,CH2][CX3](=[OX1])[NX3,NX4+][CH1,CH2][CX3](=[OX1])[O,N]”, or a glycoside substructure, defined as “[CR][OR][CHR]([OR0,NR0])[CR]”.

### 2.5. TMAP Color Gradients

The calculated properties were used to color the generated TMAP. For a clearer color gradient, some of the highest and lowest displayed values of the non-ranked properties have been adjusted. All MW values ≥1000 Da are displayed as 1,000 Da, all boiling point values ≥2000 K are displayed as 2000 K, all HBD count values ≥10 are displayed as 10, all AlogP values ≥8 are displayed as 8, all AlogP values ≤−2 are displayed as −2, and all TPSA values ≥500 are displayed as 500. The color-codes of the ranked property values were obtained by average ranking them using SciPy [39]. In average ranking, if two or more values have the same rank, the average rank of the tied values is assigned to each of them. For details on TMAP please refer to the related publication [24].

### 2.6. Support Vector Machine (SVM) and k-Nearest Neighbor (k-NN) Classifiers

The *k*-nearest neighbor (*k*-NN) algorithm is a simple ML method that predicts the query to belong to the class most found amongst its *k* nearest neighbors. A support vector machine (SVM) represents a more complex ML approach; an SVM maps its input into a high-dimensional feature space and tries to find the best separation between two classes, such as they entirely lay on the opposite side a hyperplane. To do so, the SVM maximizes the margin between the closest points, known as support vectors, and the hyperplane. Mapping features explicitly into a higher dimensional space is computationally expensive and not feasible even for small datasets. To avoid it the SVM uses the so-called “kernel trick”, which essentially uses a similarity matrix of the input data instead of the input itself; this allows the SVM to define the hyperplane and the support vectors in a less expensive manner [40]. In cheminformatics, both *k*-NN and SVM inputs can range from SMILES to various molecular descriptors. For this work, three classifiers were implemented: a MAP4 based *k*-NN (MAP4 *k*-NN), a MAP4 based SVM (MAP4 SVM), and an SVM based on physico-chemical properties (physchem SVM).

The MAP4 SVM and MAP4 *k*-NN classifiers were implemented as follows. The canonicalized SMILES without stereochemistry information used to generate the TMAP were made unique, and they were assigned to training or test set with a 50% random split. The 35 unique SMILES of the 76 entries common between both origins were randomly assigned to one origin. Both classifiers were trained using MAP4 fingerprints. In both cases, the class weights were inversely proportional to the class frequency, and their hyperparameter was optimized using a 5-fold cross-validation. During the 5-fold cross-validation, 20% of the training set was left out as a validation set, and the final set of parameters maximized the ROC AUC on the validation set. For the SVM classifier, the hyperparameter C was optimized among the values 0.1, 1, 10, 100, and 1000, resulting in C = 10. The SVM utilized a custom kernel that calculated the similarity matrix between two MAP4 fingerprints. Platt scaling [41] was used to obtain probabilistic prediction values. For the *k*-NN model, the number of nearest neighbors *k* was optimized among the values 5, 7, 9, and 11, resulting in *k* = 7. As a distance metric between two MAP4 fingerprints, the *k*-NN classifier used one minus the similarity between the two fingerprints.

The physchem SVM model was trained with the same training/test split, but using the MW, fsp3C, HBA, HBD, AlogP, TPSA, and calculated boiling point as input. The properties were scaled to zero mean and unit variance. A radial basis function (RBF) kernel [42] was used, and the hyperparameters C and γ were optimized with a grid search. C was optimized considering the values 0.1,1, 10, 100, and 1000, resulting in C= 10, and γ was optimized considering the values 0.01, 0.1, 1, 10, and 100, resulting in γ = 1.

For the evaluation of the classifiers, we considered the class “bacterium” to be the positive class and the class “fungus” to be the negative one. All SVM and the *k*-NN classifiers were implemented using scikit-learn [43], and all not mentioned hyperparameters were used in their default values. The source code for all classifiers can be found at https://github.com/reymond-group/MAP4-Chemical-Space-of-NPAtlas.

### 2.7. Classifiers Evaluation Metrics

ROC AUC is the area under the ROC curve, and the ROC curve is obtained by plotting the true positive rate (*TPR*) against the false positive rate (FPR):(2)$$TPR=\frac{TP}{TP+FP}$$
(3)$$FPR=\frac{FP}{TP+FP}$$
where *TP* stands for true positives, *TN* for true negatives, *FP* for false positives, and *FN* for false negatives predicted by the classifier.

The *F*1 score is defined as the harmonic mean of precision and recall:(4)Precision=TPR
(5)$$Recall=\frac{TP}{TP+FN}$$
(6)$$F1\;score=2\times \frac{\left(Precision\times Recall\right)}{\left(Precision+Recall\right)}$$
The balanced accuracy is defined as:(7)$$Balanced\;accuracy=\frac{TPR+\frac{TN}{TN+FN}}{2}$$
The Matthews correlation coefficient (*MCC*) is a correlation between the observed and the predicted class and it is defined as:(8)$$MCC=\frac{TP\times TN-FP\times FN}{\sqrt{\left(TP+FP\right)\left(TP+FN\right)\left(TN+FP\right)\left(TN+FN\right)}}$$
In all metrics, the probabilistic prediction values were converted into binary classification values using a threshold of 0.5.

## 3. Results and Discussion

### 3.1. The TMAP of NPAtlas

The 25,523 structures in NPAtlas were downloaded and encoded using the MAP4 fingerprint, which is well suited to analyze molecules across different sizes such as those in NPAtlas ranging between 70 and 2900 Da in MW (Table 1, Appendix A, method Section 2.1 and Section 2.2). The generated dataset was then visualized using TMAP, which represents the minimum spanning tree connecting nearest neighbors, here according to the MAP4 similarity measured as Jaccard distance (Appendix A, see method Section 2.3 for details). To understand how the NPs in NPAtlas are organized on the MAP4 TMAP, we generated color codes based on various physico-chemical descriptors, as well as on categorical classification by compound type and observed or predicted origin (Table 1, method Section 2.4 and Section 2.5, Appendix A, https://tm.gdb.tools/map4/).

Inspecting the colored TMAPs reveals that molecules are organized by structural features. For example, inspecting the TMAP colored by MW shows that most of the high MW compounds (MW ≥1000 Da, 6.8% of NPAtlas) belong to three structural families, namely peptides type compounds (minimal substructure: dipeptide), glycosides (minimal substructure: cyclic N- or O-acetal) and glycopeptides (both substructures present) (Table 2, Figure 1A and Appendix A). Typical examples of such large NPs are shown in Figure 2, featuring the cyclic peptides jizanpeptin A (NPA022688, bacterial) [44] and arbumelin (NPA020152, fungal) [45], the glycosides butirosin A (NPA009292, bacterial) [46] and quinofuracin A (NPA005440, fungal) [47], and the glycopeptides cycloaspeptide F (NPA000712, the only fungal glycopeptide in NPAtlas) [48] and orienticin D (NPA021348, bacterial) [49].

Another striking insight is provided by inspecting the TMAP colored by the fraction of sp3 carbons (fsp3C, Figure 1B), which allows the identification of areas rich in aromatic polyphenols with very low fsp3C, such as nocatrione A (NPA014210, bacterial) [50] and sydowiol E (NPA001030, fungal) [51], as well as areas populated by terpenoids with very high fsp3C such as neoverrucosane diterpenoids (e.g., neoverrucosan-5β,9β,18β-triol, NPA001820, bacterial) [52] and the anti-influenza virus diterpene wickerol B (NPA008911, fungal) [53]. The structures of these compounds are shown in Figure 2.

The TMAP not only organizes molecules by structural features, but also separates molecules according to their origin, with fungal and bacterial NPs forming well-defined groups across the TMAP (Figure 1C). This separation is striking because biosynthetic pathways in bacteria and fungi are generally similar, and because the different compound families contain NPs of both bacterial and fungal origin (Table 2).

### 3.2. Distinguishing Between Bacterial and Fungal NPs

The separation between bacterial and fungal NPs on the MAP4 TMAP and the fact that the map also separates NPs by physico-chemical descriptor values suggested to us that ML models trained either with the MAP4 fingerprint or simply with physico-chemical descriptors might be able to distinguish between NPs of bacterial or fungal origin. We investigated SVM and *k*-NN models since this type of ML models are generally well suited for classifying bioactive molecules [54]. We considered both an SVM and a *k*-NN model with MAP4, and only an SVM model with physico-chemical descriptors, and we evaluated their performance on the test set (see method Section 2.6). 

The MAP4 SVM was the best performing model with an area under the receiver operating characteristic curve (ROC AUC) of 0.97, an F1 score of 0.91, a balanced accuracy of 0.93, and a Matthews correlation coefficient (MCC) of 0.86 (Table 3). The MAP4 *k*-NN classifier also had excellent evaluation metrics with an accuracy of 0.90 and an MCC of 0.8, suggesting the high performance of the MAP4 SVM classifier might depend on a nearest neighbor effect. On the other hand, the physchem SVM performed significantly worse than the MAP4 based classifiers and was only partially capable of distinguishing between bacterial and fungal NPs (F1 score and a balanced accuracy above 0.7). This suggests that successful classification requires a model distinguishing between specific substructures and not only overall molecular properties. For closer inspection, the prediction (fungal or bacterial origin) and the performance (correct or wrong) of the best performing classifier (MAP4 SVM) are color-coded on the MAP4 TMAP of NPAtlas (Appendix A).

### 3.3. Predicting the Origin of Newly Discovered NPs

Discussion with natural product chemists informed us that assigning NPs to their origin only from its chemical structure is not trivial, and can be problematic when isolating a new NP due to the occurrence of endosymbiosis, i.e., the fact that bacteria often live as symbionts within larger organisms [51,55]. We therefore asked the question whether our MAP4 SVM classifier would correctly predict the origin of NPs newly reported in 2020 and which are not part of NPAtlas (Table 4). To our delight, the classifier correctly predicted the fungal origin for the newly reported epicospirocins 1 [56], penicimeroterpenoid A [57], and rhizolutin [58], as well as the bacterial origin of the recently reported bosamycin A [59]. The correct origin assignment is probably related to the presence of structurally similar NPs within the NPAtlas training set, illustrated here by the MAP4 nearest-neighbor NPs aspermicrone A [60], isocitreohybridone H [61], Monacolin K [62], and AIP I [63] (Figure 3).

When challenged with the recently reported NP phakefustatin A, isolated from the marine sponge *Phakellia fusca* [64] (Figure 3), the MAP4 SVM classifier predicted a bacterial origin (Table 4). Indeed, the NPAtlas training set contained closely related NPs of bacterial origin, such as the MAP4 NN Samoamide A [65] (Figure 3). Although phakefustatin A was isolated from a marine sponge, our prediction is probably correct because many marine sponges contain endosymbiotic bacteria, which can make up to 60% of the sponge biomass and are often responsible for the production of metabolites [66]. More specifically, it is known that *Phakellia fusca* coexists with diverse actinobacteria which have been held responsible for the production of many bioactive NPs found in the sponge [67].

While the example above might be a case of endosymbiosis and potential origin misclassification, it must be noted that our MAP4 SVM classifier can only label NPs as of bacterial or fungal origin. In fact, our classifier mistakenly assigns such classification to well-known non-microbial NPs (Appendix A, Appendix A). An extension of our analysis to non-microbial natural products could be of interest, however, the task cannot be completed due to a lack of annotated public datasets for NPs of diverse origins [68,69].

## 4. Conclusions

In summary, we showed that mapping the 25,523 NPs reported in NPAtlas as a MAP4 TMAP organizes molecules by physico-chemical properties and by substructures and thereby provides an unprecedented insight into the composition of this collection. Most strikingly, the map separates the different NPs according to their bacterial or fungal origin. Furthermore, a SVM model trained with the MAP4 fingerprint dataset performs remarkably well in distinguishing between fungal and bacterial NPs. The classifier can be of aid where the origin of a natural product is unknown, especially when the molecule is isolated from a symbiotic complex. The MAP4 TMAP of NPAtlas is accessible at https://tm.gdb.tools/map4/, and the source code is available at https://github.com/reymond-group/MAP4-Chemical-Space-of-NPAtlas.

## Figures and Tables

**Figure 1 biomolecules-10-01385-f001:**
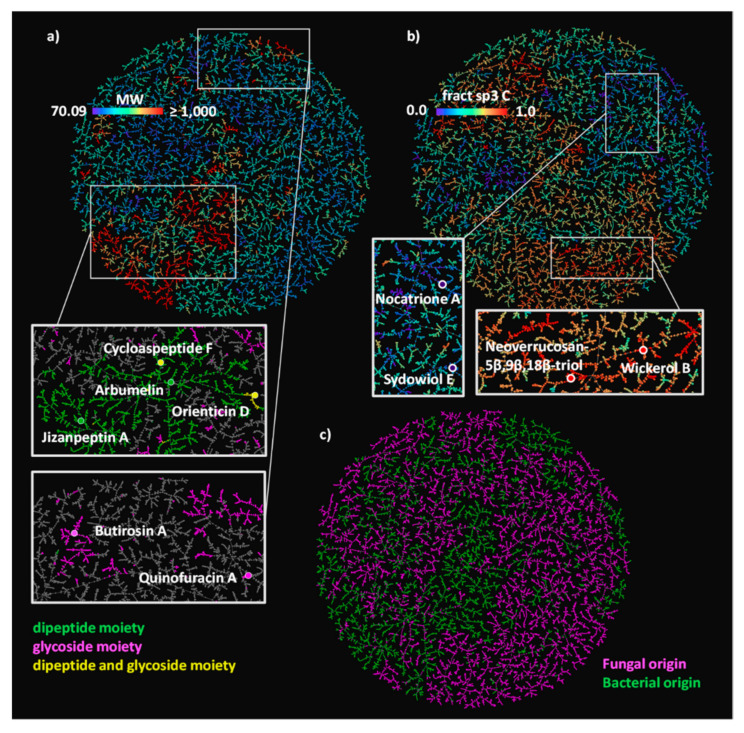
(**A**) NPAtlas MAP4 TMAP colored by MW, with a rainbow scale where the lowest values are purple, and the highest values are red. Two areas of the map are zoomed and colored by SMARTS substructure match: compounds containing a dipeptide moiety are highlighted in green, compounds containing a glycoside moiety are highlighted in magenta, compounds containing both moieties are highlighted in yellow; six examples of NPAtlas entries are reported with the same color code. (**B**) The NPAtlas MAP4 TMAP colored by fsp3C with a rainbow scale where the lowest values are purple, and the highest values are red. A low and a high fsp3C area of the map are zoomed, and two examples of polyphenols and of terpenoids are reported. (**C**) The NPAtlas MAP4 TMAP colored by a microbial origin classification, the compounds originated from fungi are colored in magenta, the compounds produced by bacteria are colored in green.

**Figure 2 biomolecules-10-01385-f002:**
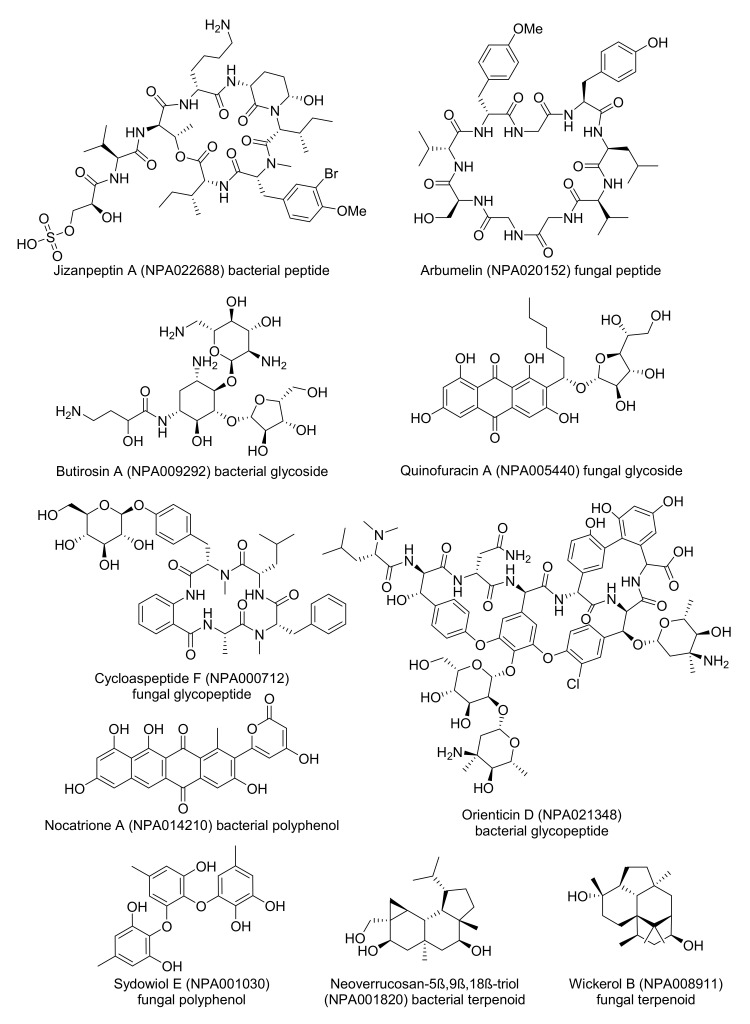
The structural formula of natural product examples selected from the TMAPs in Figure 1.

**Figure 3 biomolecules-10-01385-f003:**
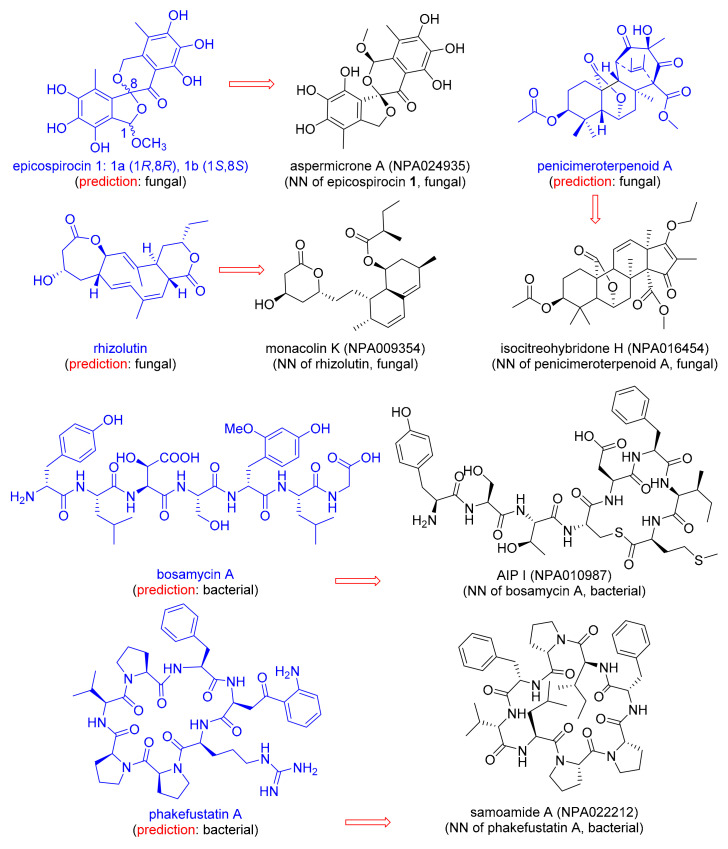
Examples of natural products reported in 2020, absent from NPAtlas, annotated with their predicted origin, and connected to its MAP4 NN in the training set.

**Table 1 biomolecules-10-01385-t001:** Calculated properties of NPAtlas molecules available as TMAP color-codes.

Property	Min. Value	Max. Value	25% Quantile	50% Quantile	75% Quantile
Molecular weight ^A^	70.1	2901.3 (1000 ^F^)	292	408.9	562.6
Sp3 C fraction ^A^	0.0	1.0	0.4	0.6	0.7
HBA count ^A,B^	0	68 (20 ^F^)	4	6	9
HBD count ^A,C^	0	47 (10 ^F^)	3	2	4
AlogP ^A,D^	−28.9 (−2 ^G^)	33.8 (8 ^F^)	1.2	2.5	4.1
TPSA ^A,E^	0.0	1135.81 (500 ^F^)	69.64	99.66	152.8
Boiling point ^A,H^	311.5	7806.5 (2000 ^F^)	890.8	1141.6	1518.5
Is Lipinski	Categorical: yes/no				
Substructures ^I^	Categorical: contains dipeptide moiety/contains glycoside moiety/contains dipeptide and glycoside moieties				
Origin	Categorical: Bacterial/Fungal				
MAP4 SVM ^J^ prediction	Categorical: Bacterial/Fungal				
MAP4 SVM ^J^ performances	Categorical: correct/wrong				

^A^ Continuous properties; shown also as rank in the map. ^B^ Hydrogen bond acceptors (HBA). ^C^ Hydrogen bond donors (HBD). ^D^ LogP Calculated following Crippen’s approach (AlogP). ^E^ topological polar surface area (TPSA). ^F^ The maximum value shown in the map, all values above are represented with the same color code. ^G^ The minimum value shown in the map, all values below are represented with the same color code. ^H^ Joback calculated boiling point. ^I^ SMARTS matched substructures. ^J^ Support vector machine (SVM).

**Table 2 biomolecules-10-01385-t002:** NPAtlas entries and unique publications number according to the origin and molecular weight.

	Fungal ^A^	Bacterial ^A^
NPAtlas entries (≥1000 Da)	15,759 (347)	9764 (1392)
Unique publications ^B^	6110 (145)	4653 (711)
Peptides (≥1000 Da) ^C^	722 (311)	2144 (901)
Glycosides (≥1000 Da) ^D^	814 (12)	1616 (421)
Glycopeptides (≥1000 Da) ^E^	1 (0)	112 (89)
Aromatic NPs (≥1000 Da) ^F^	1322 (0)	800 (31)
Aliphatic NPs (≥1000 Da) ^G^	2184 (59)	1366 (220)

^A^ Natural product origin. ^B^ Number of unique publications used for the extraction of all NPAtlas entries ^C^ Containing a dipeptide moiety. ^D^ Containing a glycoside moiety. ^E^ both glycoside and dipeptide moiety. ^F^ fsp3C < 0.2. ^G^ fsp3C > 0.8.

**Table 3 biomolecules-10-01385-t003:** SVM and *k*-NN classifier’s performance on the test set.

Classifier	ROC AUC ^A^	F1 Score ^A^	Balanced Accuracy ^A^	MCC ^A^
MAP4 SVM ^B^	0.97	0.91	0.93	0.86
MAP4 *k*-NN ^C^	0.96	0.88	0.90	0.81
Physchem SVM ^D^	0.86	0.73	0.78	0.56

^A^ Area under the receiver operating characteristic curve (ROC AUC), F1 score, balanced accuracy, and MCC are metrices used to evaluate a machine learning model. MCC can assume values from –1 to 1, all other parameters can assume values from 0 to 1, and in all cases 1 is a perfect classification. Refer to Section 2 for details. ^B^ SVM classifier trained with the MAP4 fingerprint. ^C^
*k*-NN classifier trained with the MAP4 fingerprint. ^D^ SVM trained with physiochemical properties.

**Table 4 biomolecules-10-01385-t004:** MAP4 SVM classification of new microbial natural products and of Phakefustatin A.

Natural Product	MAP4 SVM ^A^ Fungal, Bacterial	Training Set Nearest Neighbor (NN)	JD from NN ^B^
Epicospirocin 1	0.99, 0.01	Aspermicrone A (NPA024935)	0.66
Penicimeroterpenoid A	1.0, 0.0	Isocitreohybridone H (NPA016454)	0.63
Rhizolutin	0.83, 0.17	Monacolin K (NPA009354)	0.80
Bosamycin A	0.04, 0.96	AIP I (NPA010987)	0.77
Phakefustatin A	0.12, 0.88	Samoamide A (NPA022212)	0.68

^A^ Predicted origin: fungal or bacterial. ^B^ Approximated Jaccard distance (JD), see Section 2 for details from the training set NN.

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
