# Peer review of "Assigning the Origin of Microbial Natural Products by Chemical Space Map and Machine Learning"

_biomolecules, 2020, doi:10.3390/biom10101385_

Round 1

Reviewer 1 Report

The manuscript by Capecchi and Reymond provides a nice analysis of microbial natural products from the Natural Products Atlas including interactive visualization maps, as well as results in a machine learning classifier to distinguish NPs from bacterial or fungal origin. I only have a few comments:

  1. It would be good to emphasize the importance/usage of assigning the origin of microbial natural products
  2. Page 2, line 63: “The NPAtlas counts 23,928 unique SMILES and 76 entries common among both origins”. It seems the authors didn’t mention how they handled the SMILES with two origins for the machine learning classifiers. Also, can different SMILES have the same MAP4 fingerprint? If so, there will be conflict labels of one fingerprint.  
  3. Section 3.2 and Table 3, the authors should indicate the performance is on which data set.
  4. In Abstract, “The resulting interactive map (https://tm.gdb.tools/map4/npatlas_map_tmap/) organizes molecules by physico-chemical properties and compound families such as peptides, glycosides, polyphenols or terpenoids.” In the interactive map I didn’t find options for polyphenols or terpenoids. The peptides and glycosides were identified by SMARTS patterns and were shown in colors on the map, but the areas rich in polyphenols or terpenoids were identified by the fraction of sp3 carbons (fsp3C) and this would not be clear which nodes are polyphenols or terpenoids. 
  5. Some references have formatting issue, e.g. 45, 50, 58

Author Response

Reviewer 1

The manuscript by Capecchi and Reymond provides a nice analysis of microbial natural products from the Natural Products Atlas including interactive visualization maps, as well as results in a machine learning classifier to distinguish NPs from bacterial or fungal origin. I only have a few comments:

  • It would be good to emphasize the importance/usage of assigning the origin of microbial natural products

Our answer: We have stated more clearly in the conclusions “The classifier can be of aid where the origin of a natural product is unknown, especially when the molecule is isolated from a symbiotic complex”.

  • Page 2, line 63: “The NPAtlas counts 23,928 unique SMILES and 76 entries common among both origins”. It seems the authors didn’t mention how they handled the SMILES with two origins for the machine learning classifiers. Also, can different SMILES have the same MAP4 fingerprint? If so, there will be conflict labels of one fingerprint.

Our answer: Thanks for pointing this out. The 35 unique SMILES corresponding to the 76 entries were randomly assigned to one of the two origins. The related method section has been extended as follows: “The canonicalized SMILES without stereochemistry information used to generate the TMAP were made unique, and they were assigned to training or test set with a 50% random split. The 35 unique SMILES of the 76 entries common between both origins were randomly assigned to one origin. While the non-hashed version of the fingerprint is very unlikely to be the same for two different molecules, the dimensionalities reduction using MinHashing can introduce feature collision. However, this is not different than the modulo operation of ECFP and it can occur also with novel compounds of which we want to predict the origin. We did not check for conflicting labels across same fingerprint values, and we don’t consider it to be an issue.

  • Section 3.2 and Table 3, the authors should indicate the performance is on which data set.

Our answer: We have extended the sentence “We considered both an SVM and a k-NN model with MAP4, and only an SVM model with physico-chemical descriptors and we evaluated their performance on the test set (see method section 2.6).” in section 3.2,  and we have changed the caption of Table 3 to “SVM and k-NN classifiers performance on the test set”.

  • In Abstract, “The resulting interactive map (https://tm.gdb.tools/map4/npatlas_map_tmap/) organizes molecules by physico-chemical properties and compound families such as peptides, glycosides, polyphenols or terpenoids.” In the interactive map I didn’t find options for polyphenols or terpenoids. The peptides and glycosides were identified by SMARTS patterns and were shown in colors on the map, but the areas rich in polyphenols or terpenoids were identified by the fraction of sp3 carbons (fsp3C) and this would not be clear which nodes are polyphenols or terpenoids.

Our answer: We agree with the reviewer, even if the fsp3C color code is helpful to identify areas on the map reach in polyphenols and terpenoids (as described in section 3), the sentence in the abstract is misleading and it has been changed to “The resulting interactive map (https://tm.gdb.tools/map4/npatlas_map_tmap/) organizes molecules by physico-chemical properties and compound families such as peptides and glycosides“.

  • Some references have formatting issue, e.g. 45, 50, 58

Our answer: Thanks, the three references (now 49, 54, and 62) have been fixed.

Reviewer 2 Report

The authors have found a reliable method to distinguish the bacterial or fungi origins of any molecules. Therefore, the applicability of this work has a large range, for example, in molecular bio prospecting  analysis. In my opinion, this paper is well written, has a great scientific soundness and an extraordinary scientific potential. I am confident that this work will open the door to other work and employing in the design and the discovery of new compound. Therefore, I strongly recommend its publishing in this form.

Much appreciated the opportunity,

Sincerely yours, 

Author Response

Reviewer 2

The authors have found a reliable method to distinguish the bacterial or fungi origins of any molecules. Therefore, the applicability of this work has a large range, for example, in molecular bio prospecting analysis. In my opinion, this paper is well written, has a great scientific soundness and an extraordinary scientific potential. I am confident that this work will open the door to other work and employing in the design and the discovery of new compound. Therefore, I strongly recommend its publishing in this form.

Much appreciated the opportunity,

Sincerely yours,

Our answer: Thank you very much for your feedback and appreciation.

Reviewer 3 Report

Assigning the origin of microbial natural products by chemical space map and machine learning

This paper introduced an interesting fingerprint called MinHashed Atom Pair fingerprint with a diameter of four bonds (MAP4) to characterize the structure similarity of natural products, especially microbial natural products. More importantly, using this fingerprint, they trained a machine learning model to distinguish the NPs from bacteria to fungal. The whole paper is well written and organized clearly.

However, we have some major and minor comments for improvement.

Major comments:

  1.  
  2. Recent machine learning applications on the study of traditional Chinese herb medicine should be also mentioned in the Introduction e.g. https://pubmed.ncbi.nlm.nih.gov/31765369/; https://pubmed.ncbi.nlm.nih.gov/32869826/.
  3. In addition to TMAP, TSNE is another commonly used tool for visualization of high-dimension datasets. What the result of TSNE looks like compared to TMAP?
  4. Line 72-74: it is unclear the difference between value and index. Should they refer to the same thing? Also, since the denominator is ‘the number of elements of fingerprint a’, does it mean that the similarity is asymmetric, i.e. s(a,b) is not equal to s(b,a)?
  5. Line 82-84: Why the TMAP algorithm generates a forest of 32 trees? It the number of trees a parameter of the TMAP algorithm which can be adjusted? Similar comments are for the rational of the choice of 20 nearest neighbors.
  6. Line 218, machine learning models are able to distinguish between NPs of bacterial or fungal origin by either the MAP4 fingerprint or simply the physico-chemical descriptors. It will be interesting if the author can explore further which fingerprint features or physico-chemical descriptors contribute to the prediction accuracy the most.

Minor comments:

  1. Line 41-43 is lack of references
  2. Line 72: it is unclear the difference between value and index. Should they refer to the same thing?
  3. Line 42-48: The authors mentioned that ECFP4 performs poorly in NP prediction and MAP4 is a better choice. However, they did not show the performance of ECFP4 in predicting the microbial NP. We recommend the authors to include other types of fingerprints and train machine learning model as the reference.

Author Response

Reviewer 3

This paper introduced an interesting fingerprint called MinHashed Atom Pair fingerprint with a diameter of four bonds (MAP4) to characterize the structure similarity of natural products, especially microbial natural products. More importantly, using this fingerprint, they trained a machine learning model to distinguish the NPs from bacteria to fungal. The whole paper is well written and organized clearly.

However, we have some major and minor comments for improvement.

Major comments:

  • Recent machine learning applications on the study of traditional Chinese herb medicine should be also mentioned in the Introduction e.g. https://pubmed.ncbi.nlm.nih.gov/31765369/; https://pubmed.ncbi.nlm.nih.gov/32869826/.

Our answer: We have extended the introduction as follows: “Furthermore, machine learning (ML) has been extensively applied to natural product structures, for example to classify limonoids and protolimonoids [6], to establish the structural class of a natural product with its NMR data [7], to learn estimates of natural product conformational energies [8], to generate derivates of NPs or compounds with natural product characteristics [9–11], to predict Meridian in Chinese traditional medicine [12], and to elucidate the biological effects of natural products [13]” and added the references proposed by the reviewer (12, 13).

  • In addition to TMAP, TSNE is another commonly used tool for visualization of high-dimension datasets. What the result of TSNE looks like compared to TMAP?

Our answer: In our recent publication “Visualization of very large high-dimensional data sets as minimum spanning trees”, we showed that TMAP works better than other dimensionality reduction methods such as t-SNE. However, the reviewer is correct that the comparison should be mentioned also here. We have extended the introduction with the following sentence: “TMAP performs better for the visualization of large high-dimensional data sets than other dimensionality reduction methods such as t-SNE [25] or UMAP [26] . Furthermore, TMAP is particularly well suited to analyze databases of molecules associated with MinHashed fingerprints”.

  • Line 72-74: it is unclear the difference between value and index. Should they refer to the same thing? Also, since the denominator is ‘the number of elements of fingerprint a’, does it mean that the similarity is asymmetric, i.e. s(a,b) is not equal to s(b,a)?

Our answer: The MAP4 fingerprint is an array of unsorted numbers, each number in the array is characterized by its value and its position in the array (index). The number of values in a MAP4 fingerprint (its length) depends on its dimensionality; therefore, when comparing two MPA4 fingerprints with the same dimensionality the similarity is symmetric. We have extended the section as follows: “The resulting set of strings is hashed to integers using the SHA-1 algorithm [27] and MinHash scheme [28]. The obtained MAP4 fingerprint is an array of unsorted numbers, where each feature is characterized by its value and its position in the array (index). “

  • Line 82-84: Why the TMAP algorithm generates a forest of 32 trees? It the number of trees a parameter of the TMAP algorithm which can be adjusted? Similar comments are for the rational of the choice of 20 nearest neighbors.

Our answer: The number of trees in the LSH forest and the number of nearest neighbors used to generate the map are both parameters of the TMAP method. Since they are not the default ones and they influence the compounds connectivity in the final TMAP, we think it is important to specify them. To clarify, we have modified the section as follows: “In short, the indices generated by the MinHash procedure of the MAP4 calculation are used to create a locality-sensitive hashing (LSH) forest [29] of n trees. For each NPAtlas entry, the k approximate nearest neighbors (NNs) in the MAP4 feature space are then extracted from the LSH forest to form a graph in which nodes are the structures and edges are the NN relationships weighted by the fingerprint distance. The Kruskal’s algorithm is then applied to remove cycles and to find the path with the lowest total distance between all molecules in the graph [30]. Finally, Fearun [31] is used to interactively display the obtained minimum spanning tree. In this this study we set n = 32 and k = 20.”

  • Line 218, machine learning models are able to distinguish between NPs of bacterial or fungal origin by either the MAP4 fingerprint or simply the physico-chemical descriptors. It will be interesting if the author can explore further which fingerprint features or physico-chemical descriptors contribute to the prediction accuracy the most.

Our answer: Thank you for this suggestion. Although an analysis of the MPA4 fingerprint features that most contribute to the prediction could be of interest, in the current implementation of MAP4, it is not possible to directly trace back the original atom pair shingle from the MinHashed feature.

Minor comments:

  • Line 41-43 is lack of references

Our answer: We have added a reference to figure S1 and to the MAP4 paper.

  • Line 72: it is unclear the difference between value and index. Should they refer to the same thing?

Our answer: The MAP4 fingerprint is an array of unsorted numbers, each number in the array is characterized by its value and its position in the array (index). We have extended the section as follows: “The resulting set of strings is hashed to integers using the SHA-1 algorithm [27] and MinHash scheme [28]. The obtained MAP4 fingerprint is an array of unsorted numbers, where each feature is characterized by its value and its position in the array (index). “

  • Line 42-48: The authors mentioned that ECFP4 performs poorly in NP prediction and MAP4 is a better choice. However, they did not show the performance of ECFP4 in predicting the microbial NP. We recommend the authors to include other types of fingerprints and train machine learning model as the reference.

Our answer: The poorer performance of ECFP4 compared to MAP4 for large NP-like molecules such as lipids, oligosaccharides and peptides is described in detail in our recent publication on MAP4, ref. no. 17 in our manuscript. We report the present study on microbial natural products as an implementation of our MAP4 fingerprint of interest for natural products. A comparison of different molecular fingerprints for their performance in analyzing natural products goes beyond the scope of the present report.

Round 2

Reviewer 3 Report

The authors have answered my comments.